# THE GEOMETRY OF SELF-SUPERVISED LEARNING MODELS AND ITS IMPACT ON TRANSFER LEARNING

## ABSTRACT

Self-supervised learning (SSL) has emerged as a desirable paradigm in computer vision due to the inability of supervised models to learn representations that can generalize in domains with limited labels. The recent popularity of SSL has led to the development of several models that make use of diverse training strategies, architectures, and data augmentation policies with no existing unified framework to study or assess their effectiveness in transfer learning. We propose a data-driven geometric strategy to analyze different SSL models using local neighborhoods in the feature space induced by each. Unlike existing approaches that consider mathematical approximations of the parameters, individual components, or optimization landscape, our work aims to explore the geometric properties of the representation manifolds learned by SSL models. Our proposed manifold graph metrics (MGMs) provide insights into the geometric similarities and differences between available SSL models, their invariances with respect to specific augmentations, and their performances on transfer learning tasks. Our key findings are two fold: $(i)$ contrary to popular belief, the geometry of SSL models is not tied to its training paradigm (contrastive, non-contrastive, and cluster-based); $(ii)$ we can predict the transfer learning capability for a specific model based on the geometric properties of its semantic and augmentation manifolds.

## 1 INTRODUCTION

Self-supervised learning (SSL) for computer vision applications has empowered deep neural networks (DNNs) to learn meaningful representations of images from unlabeled data (1; 2; 3; 4; 5; 6; 7; 8; 5; 9; 10; 11). These methods learn a feature space embedding that is invariant to data augmentations (e.g., cropping, translation, color jitter) by maximizing the agreement between representations from different augmentations of the same image. The resulting models are then used as general-purpose feature extractors and have been shown to achieve better performance for transfer learning as compared to features obtained from a supervised model (12). Broadly, SSL models can be categorized into contrastive (13; 2), non-contrastive (14; 15), prototype/clustering (16; 17). There exist multiple differences in the resulting models, even among models belonging to the same category. For example, popular SSL models available as pre-trained networks (18) can differ in terms of training parameters (loss function, optimizer, learning rate), architecture (DNN backbone, projection head, momentum encoder), model initialization (weights, batch-normalization parameters, learning rate schedule), etc.

Recently, researchers have focused on developing an understanding of specific components of SSL models by studying the loss function used to train the models and its impact on the learned representations. For instance, (19) analyzes contrastive loss functions and the dimensional collapse problem. (20) also analyzes contrastive losses and describes the effect of augmentation strength as well as the importance of non-linear projection head. (21) quantifies the importance of data augmentations in a contrastive SSL model via a distance-based approach. (22) explores a graph formulation of contrastive loss functions with generalization guarantees on the representations learned. In (23), the importance of the temperature parameter used in the SSL loss function and its impact on learning is examined. (24) performs a spectral analysis of DNN's mapping induced by non-contrastive loss and the momentum encoder approach. However, these studies are unable to provide a *unified analysis* of the myriad of existing SSL models. Besides, these theoretical approaches only provide insights into the embedding obtained *after* the projection head, while in practice it is the mapping provided by the encoder that is actually used for transfer learning.

Closer to our approach, the general transfer performance of an SSL model has been predicted based on the performance it can achieve on the ImageNet dataset (25). This idea was shown to be effective for transfer datasets that are similar to ImageNet, but it cannot be generalized to all transfer learning problems (12). In fact, if ImageNet performance were highly correlated to general transfer learning performance, then it is unclear why SSL models would be needed, as one could simply use supervised training with ImageNet to obtain image representations for transfer learning. Furthermore, existing empirical evaluations such as (25) only provide a *somewhat coarse and partial understanding* of SSL models. For example, they do not provide insights into how the level of invariance to specific augmentation in an SSL model relates to its performance on a given downstream task. Since it has been observed that invariance to some augmentations can be beneficial in some cases and harmful in others (26), our goal is to develop a more direct and quantitative understanding of augmentation invariance and how this invariance determines performance.

To achieve this goal, we propose a *geometric* perspective to understand SSL models and their transfer capabilities. Our approach analyzes the manifold properties of SSL models by using a data-driven graph-based method to characterize the geometry of data and their augmentations, as illustrated on the left of Figure 1. Specifically, we develop a set of **manifold graph metrics** (MGMs) to quantify the geometric properties of existing SSL models. This allows us to provide insights about the similarities and differences between models (Figure 1, right) and link their ability to transfer to specific characteristics of the target task. Because our approach can be applied directly to sample data points and their augmentations, it has several important advantages. First, it is agnostic to specific training procedures, architecture, and loss functions; Second, it can be applied to the data embeddings obtained at any layer of the SSL model, thus alleviating the challenge induced by the presence of projection heads; Third, it enables us to compare different feature representations of the same data point, even if these representations have different dimensions.

We are interested in using our approach to answer the following questions about SSL models:
$(i)$ **What are the geometric differences between the feature spaces of various SSL models?**
$(ii)$ **What geometric properties allow for better transfer learning capability for a specific task?**

For our transfer study, we perform our MGM evaluation on 14 SSL models under 5 augmentation setting and show their impact in 8 downstream tasks comprising of 18 datasets in total.

Our contributions are summarized as follows:

- We develop quantitative tools (i.e., MGMs) capable of capturing important geometrical aspects of SSL representations, such as the degree of equivariance-invariance, the curvature, and the intrinsic dimensionality, Sec. 3.
- We leverage the proposed MGMs to explore the geometric differences and similarities between SSL models. As illustrated in the right part of Figure 1, we show that SSL models can be clustered using these geometric properties into three main groups that are not entirely aligned with the paradigm upon which they were trained, Sec. 4.
- We analyze the geometric differences between a Vision Transformer (ViT) and a convolutional network (ResNet). We show that while ResNet is biased towards a collapsed representation at initialization, ViTs are not. This initialization bias leads to different geometrical behavior (attraction/repulsion of representations) between the two architectures when training under an SSL regime as detailed in Sec. 4.
- We demonstrate that observed MGMs are a strong indicator of the transfer learning capabilities of SSL models for most downstream tasks. Therefore showing that specific geometrical properties are crucial for a given transfer learning task, Sec. 5.

## 2 BACKGROUND

**Self-supervised Learning:** SSL models are generated by producing multiple versions of the same image, via data augmentation, and training a DNN such that their embeddings coincide. Many SSL models are obtained with architectures that cascade two networks, the backbone encoder, from which the representation to be used for a downstream task is extracted, and the projection head, from which the output is fed into the SSL loss function. The main risk in such an approach is the so-called the feature collapse phenomenon (19; 24; 27), where the learned representations are invariant to input samples that belong to different manifolds. To reduce the risk of feature collapse, multiple SSL

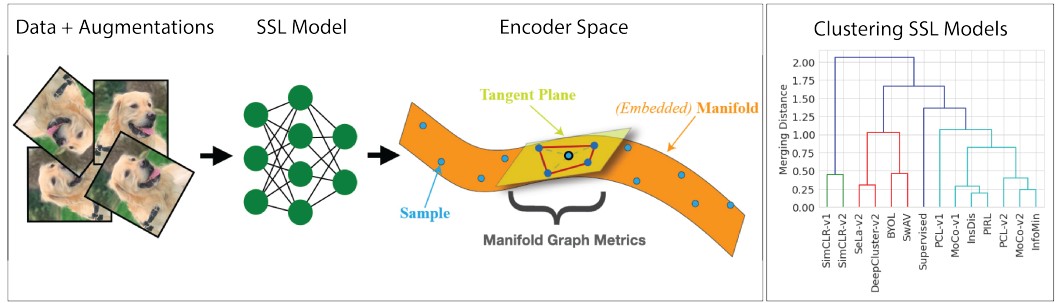

Figure 1: (**Left**) **Approach toward the analysis of SSL models** For each model, we use as input of the DNN the validation set of ImageNet as well as their augmented versions. The output of the backbone encoder is used to quantify the properties of the manifold induced by the SSL algorithm. Specifically, we develop **Manifold Graph Metrics** (Sec. 3) that capture manifold properties known to be crucial for transfer learning. The MGMs allow us to capture the specificity of each SSL model (Sec. 4) and to characterize their transfer learning capability (Sec. 5). (**Right**) We provide the dendrogram of the SSL models considered in this paper based on the manifold graph metrics we propose. Although the underlying hyper-parameters, loss functions, and SSL paradigms are different, the manifolds induced by the SSL algorithms can be categorized into three types. An important observation is that the resulting clusters are not necessarily aligned with the different classes of SSL algorithms. This result shows that although some training procedures appear to be more similar, one is required to provide a deeper analysis to understand in what aspect SSL models differ.

algorithms have been proposed. Contrastive SSL methods (e.g., simCLR-v1 (13), simCLR-v2 (2), MoCo-v1 (28), MoCo-v2 (29), Infomin (11), PIRL (30), InsDis (31)) make use of negative pairs to force the embedding of data points in an unaugmented pairs to be pushed away during training. Non-contrastive SSL methods (BYOL (4), DINO (32)) utilize a teacher-student based approach where the weights of the students are the exponential moving average of the teacher's weights. Protopype/clustering SSL methods (SeLA-v2 (33), DeepCluster-v2 (17), SwAV (17), PCL-v1 and PCL-v2 (16)) enforce consistency obtained from different transformations of the same image by comparing cluster assignments without negative pair comparisons.

We will analyze the geometrical differences of these SSL models and the impact of geometry on the transfer learning capability. To perform this analysis, we will use pre-trained SSL models based on ResNet50(1x) backbone encoder architecture as well as a supervised model available with the Pytorch library (34). We will also compare the geometrical differences between a ResNet50 and ViT architecture both at initialization and after SSL training using DINO loss (32).

**Graphs:** Graph and local neighborhood methods have played a significant role in machine learning tasks such as manifold learning (35), semi-supervised learning (36; 37), and, more recently, graph-based analysis of deep learning (38; 39; 40). Typically their use in this context is motivated by their ability to represent data with irregular positions in space $(\boldsymbol{x}_i)_{i=1}^N$ rather than directly modeling the distribution of the data $P(\boldsymbol{x})$. This type of DNN analysis starts by constructing good neighborhood representation (41; 42), which is also a crucial first step in our framework. The most common approaches to defining a neighborhood are k-nearest neighbor (kNN) and $\epsilon$-neighborhood. However, these approaches select points in a neighborhood based on only distance to the query point, and do not consider their relative positions, while also relying on ad hoc procedures to select parameter values (e.g., $k$ or $\epsilon$). For this reason, we make use of non-negative kernel regression (NNK) (43) to define neighborhoods and graphs for our manifold analysis. Unlike kNN, which can be seen as thresholding approximation, NNK can be interpreted as a form of basis pursuit (44), which leads to better neighborhood construction with robust local estimation performance in several machine learning tasks (45; 46). Of particular importance for our proposed SSL analysis framework is the fact that NNK neighborhood have a geometric interpretation. While in kNN points $\boldsymbol{x}_j$ and $\boldsymbol{x}_k$ are included in the neighborhood of a data point $\boldsymbol{x}_i$, denoted by $\mathcal{N}(\boldsymbol{x}_i) = \{\boldsymbol{x}_{i_1}, \ldots, \boldsymbol{x}_{i_{n_i}}\}$, solely based on their distances to $\boldsymbol{x}_i$, i.e., $d(\boldsymbol{x}_i, \boldsymbol{x}_j)$ and $d(\boldsymbol{x}_i, \boldsymbol{x}_k)$, in NNK this decision is made by also taking into account $d(\boldsymbol{x}_j, \boldsymbol{x}_k)$, so that $\boldsymbol{x}_j$ and $\boldsymbol{x}_k$ are both included in $\mathcal{N}(\boldsymbol{x}_i)$ only if they are not geometrically *redundant*. As a result, an NNK neighborhood can be described as a *convex polytope*

*approximation* of $\boldsymbol{x}_i$, denoted $\mathcal{P}(\boldsymbol{x}_i)$, the size and shape of which is determined by the local geometry of the data available around $\boldsymbol{x}_i$ (43). This geometric property is particularly important for data that is not uniformly sampled and lies on a lower dimensional manifold in high dimensional vector space, which is typical for feature embeddings in DNN models. Note that NNK uses kNN as an initialization step, after which it has only modest additional runtime requirement (43). Thus, we can scale our analysis to large datasets by speeding up the initialization using computational tools developed for kNN (47; 48). NNK requires kernels with a range in $[0, 1]$. In this work, we use the cosine kernel since the encoder representations are obtained after *ReLU* and so the kernel satisfies the requirement.

## 3 MANIFOLD GRAPH METRICS

We use the NNK polytope $\mathcal{P}(\boldsymbol{x}_i)$ to characterize the neighborhood of $\boldsymbol{x}_i$ and the local and global geometry of the data manifolds. Our approach, summarized in Figure 1, is motivated by the observation that while the DNNs used in SSL involve complex non-linear mappings, the induced transformations and the structure of the representation space can be inferred from the data samples by observing their relative positions in that space. In particular, we propose a set of **Manifold Graph Metrics (MGMs)** that provide intuition and quantitative data to characterize the geometry of an SSL model, Figure 2. We focus on three manifold properties that we consider important for the understanding of SSL-based feature embeddings and their transfer capabilities; $(i)$ the level of invariance (or equivariance) with respect to a given transformation or augmentation, $(ii)$ the curvature of the manifold, $(iii)$ the local intrinsic dimension of the manifold.

**Invariance-Equivariance Metric:** Given input images $\{\boldsymbol{x}_i\}_{i=1}^N$, we apply the NNK neighborhood definition to the feature space representations $(\boldsymbol{f}(\boldsymbol{x}_i))$ of the images to obtain $N$ convex polytopes. We define the **diameter** of an NNK polytope as the maximum distance between the nodes forming the polytope, i.e.,

$$\text{Diam}(\mathcal{P}(\boldsymbol{f}(\boldsymbol{x}_i))) = \max_{k,l \in \mathcal{N}(\boldsymbol{f}(\boldsymbol{x}_i)))} \left\| \hat{\boldsymbol{f}}(\boldsymbol{x}_k) - \hat{\boldsymbol{f}}(\boldsymbol{x}_l)) \right\|_2 \tag{1}$$

Figure 2: **MGMs in feature space are based on NNK polytopes**. We display data samples (red and blue dots) and two NNK polytopes $(\mathcal{P}(\boldsymbol{x}_i), \mathcal{P}(\boldsymbol{x}_j))$ in the encoder space that approximate the underlying manifold of the SSL model (gray surface). Our proposed MGMs capture invariance (polytope diameter), manifold curvature (angle between neighboring polytopes), and local intrinsic dimension (number of vertices in polytope) of the output manifold for a given SSL model.

where $\hat{\boldsymbol{f}}$ corresponds to the $l_2$-normalized feature embedding of a given input. These polytope diameters take values in the range $[0, 2]$ and provide a quantitative measure of how much the input samples have been contracted or dilated by the DNN backbone of the SSL model. Thus, a constant or collapsed mapping, where multiple $\boldsymbol{x}_i$ are mapped to the same $\boldsymbol{f}$, would lead to a degenerate polytope with diameter equal to zero. Instead, a diameter of 2 corresponds to a mapping where the neighbors are maximally scattered. Now, by considering as input only the augmented versions of an image, the diameter of the NNK polytope captures the level of invariance-equivariance of the DNN with respect to this specific augmentation. Alternatively, if we constrain ourselves to input images that have the same class label, the polytope diameter indicates the level of invariance-equivariance of the representation to samples that belong to that specific class. In this setting, a lower value of diameter corresponds to invariance of the DNN mapping to that transformation while higher values corresponding to equivariance.

**Curvature Metric:** We study the curvature of a manifold by comparing the orientations of NNK polytopes corresponding to neighboring input samples. That is, given two samples $\boldsymbol{x}_i$ and $\boldsymbol{x}_j$ that are NNK neighbors, and their respective polytopes $\mathcal{P}(\boldsymbol{f}(\boldsymbol{x}_i)), \mathcal{P}(\boldsymbol{f}(\boldsymbol{x}_j))$, we evaluate the angle between the subspace spanned by the neighbors $\mathcal{N}(\boldsymbol{f}(\boldsymbol{x}_j)), \mathcal{N}(\boldsymbol{f}(\boldsymbol{x}_i))$ that make up the polytopes. Concretely,

we use the concept of affinity between subspaces (49) to define the affinity between two polytopes as a quantity in the interval $[0, 1]$ given by

$$\text{Aff}(\mathcal{N}(\boldsymbol{f}(\boldsymbol{x}_i)), \mathcal{N}(\boldsymbol{f}(\boldsymbol{x}_j))) = \sqrt{\frac{\cos^2(\theta_1) + \cdots + \cos^2(\theta_{n_i n_j})}{n_i n_j}}, \tag{2}$$

where $\theta_k$, $k = 1, \ldots, n_j n_i$ are the principal angles between the two subspaces spanned by the vectors $\mathcal{N}(\boldsymbol{f}(\boldsymbol{x}_j))$ and $\mathcal{N}(\boldsymbol{f}(\boldsymbol{x}_i))$, containing all the NNK neighbors of $\boldsymbol{x}_j$ and $\boldsymbol{x}_i$, respectively. Intuitively, this metric is equal to zero when the subspaces are orthogonal (polytopes oriented in perpendicular directions), while a value of one corresponds to zero curvature or subspace alignment.

**Intrinsic Dimension Metric:** The local intrinsic dimension of a manifold can be estimated as the number of neighbors selected by NNK. It was shown in (43; 50), that the number of neighbors per polytope, i.e., $n_i$, correlates with the local dimension of the manifold around a data point $i$. This observation is consistent with geometric intuitions; the number of NNK neighbors is decided based on the availability of data that span orthogonal directions, i.e., the local subspace of manifold:

$$\text{ID}(\boldsymbol{f}(\boldsymbol{x}_i)) = \text{Card}(\mathcal{N}(\boldsymbol{f}(\boldsymbol{x}_i))), \tag{3}$$

where Card denotes the cardinal operator.

**Experimental settings:** For all SSL models, we analyze their equivariance-invariance, subspace curvature, and intrinsic dimension in a set of controlled and interpretable experiments. While in this work, we consider as inputs the validation set of ImageNet (51), the framework is applicable to any dataset (with or without labels). Our experimental setup is as follows: for each input sample, $(i)$ select an augmentation setting, $(ii)$ sample $T = 50$ augmentations of the image, $(iii)$ compute the similarity graph using NNK neighborhoods, and $(iv)$ extract the proposed MGMs (Sec.3). We limit ourselves to 5 augmentation types: $(i)$ *Semantic augmentations (Sem.)* , where we consider all the samples belonging to each class as augmented versions of each other in the semantic direction of the manifold, $(ii)$ *Augmentations (Augs.)* which corresponds to the sequential application of various augmentations used during most SSL training process (random horizontal flip, colorjitter, random grayscale, Gaussian blur, and random cropping), $(iii)$ *Crop* and $(iv)$ *Colorjitter (Colorjit.)*, specific augmentations that are part of the augmentation policies used to train the SSL, $(v)$ *Rotate*, an augmentation that was not used for training SSL but is considered important for some transfer tasks.

Therefore, for each sample and for each MGM we obtain a value (local manifold analysis), while the collection of these per-sample MGMs for the entire validation set gives a distribution (such as the one displayed in Figure 3). In order to extract differences between SSL models and highlight which geometric properties favor specific transfer learning tasks, we extract two statistics from these distributions of MGMs (global manifold analysis): the mean (denoted by the metric name) and the spread (referred to as metric spread). In total, for each SSL model, we obtain 26 geometric features, namely, {Sem., Augs., Crop, Colorjit., Rotate} $\times$ { Equivariance, Equivariance spread, Affinity, Affinity spread, Nb. of neighbors }, as well as the affinity and spread between Sem.-Augs, namely, { Sem.-Augs. Affinity, Sem.-Augs Affinity spread }. Additional details about each metric and their variability across all models is given in Appendix C.

In Sec. 5, we evaluate the capability of the MGMs to characterize the transfer learning capability of SSL models. While we highlight here the overall setting of the transfer learning task, details regarding the datasets can be found in Appendix I and regarding the transfer learning training settings in (12): *FewShotKornblith* and *ManyShotLinear* correspond to few/many-shot classification using SSL features extracted on 11 image classification datasets (25); *ManyShotFinetune* is related to the classification performance as in the previous setting, but with entire network along with the feature extractor updated; *FewShotCDFSL* corresponds to few-shot transfer performance in cross-domain image datasets such as CropDiseases, EuroSAT, ISIC, and ChestX datasets (52); *DetectionFinetune* and *DetectionFrozen* refer to object detection task evaluated on PASCAL VOC dataset (53); *DenseSNE* corresponds to dense surface normal estimation evaluated on NYUv2 (54); and *DenseSeg* refers to dense segmentation task evaluated on ADE20K dataset (55).

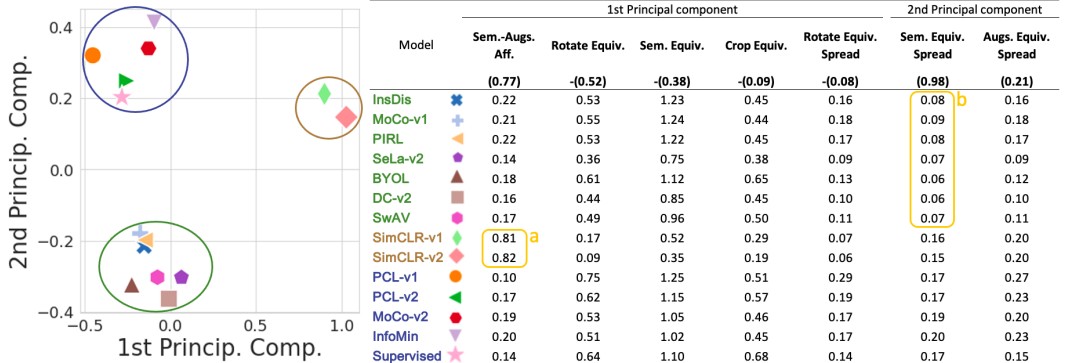

| Model | 1st Principal component | | | | | 2nd Principal component | |
|---|---|---|---|---|---|---|---|
| | Sem.-Augs. Aff. | Rotate Equiv. | Sem. Equiv. | Crop Equiv. | Rotate Equiv. Spread | Sem. Equiv. Spread | Augs. Equiv. Spread |
| | (0.77) | -(0.52) | -(0.38) | -(0.09) | -(0.08) | (0.98) | (0.21) |
| InsDis | 0.22 | 0.53 | 1.23 | 0.45 | 0.16 | 0.08 | 0.16 |
| MoCo-v1 | 0.21 | 0.55 | 1.24 | 0.44 | 0.18 | 0.09 | 0.18 |
| PIRL | 0.22 | 0.53 | 1.22 | 0.45 | 0.17 | 0.08 | 0.17 |
| SeLa-v2 | 0.14 | 0.36 | 0.75 | 0.38 | 0.09 | 0.07 | 0.09 |
| BYOL | 0.18 | 0.61 | 1.12 | 0.65 | 0.13 | 0.06 | 0.12 |
| DC-v2 | 0.16 | 0.44 | 0.85 | 0.45 | 0.10 | 0.06 | 0.10 |
| SwAV | 0.17 | 0.49 | 0.96 | 0.50 | 0.11 | 0.07 | 0.11 |
| SimCLR-v1 | 0.81 | 0.17 | 0.52 | 0.29 | 0.07 | 0.16 | 0.20 |
| SimCLR-v2 | 0.82 | 0.09 | 0.35 | 0.19 | 0.06 | 0.15 | 0.20 |
| PCL-v1 | 0.10 | 0.75 | 1.25 | 0.51 | 0.29 | 0.17 | 0.27 |
| PCL-v2 | 0.17 | 0.62 | 1.15 | 0.57 | 0.19 | 0.17 | 0.23 |
| MoCo-v2 | 0.19 | 0.53 | 1.05 | 0.46 | 0.17 | 0.19 | 0.20 |
| InfoMin | 0.20 | 0.51 | 1.02 | 0.45 | 0.17 | 0.20 | 0.23 |
| Supervised | 0.14 | 0.64 | 1.10 | 0.68 | 0.14 | 0.17 | 0.15 |

Table 1: (**Left**) **Projection of SSL models MGMs onto principal components.** We observe three distinct clusters based on the observed MGMs that distinguishes the geometric similarities and differences between the various models. Note that these clusters are not necessarily aligned with the underlying SSL training paradigm, i.e., contrastive, non-contrastive, prototype-clustering based. (**Right**) **MGMs that make up the principal components** and their values for each model, where we indicate by the two yellow boxes (a) and (b) the MGMs that capture the maximum variation of the models along the principal directions.

## 4 GEOMETRY OF SSL MODELS

In this section, we aim to characterize the geometric properties of the manifolds of various SSL models, as a way to highlight the differences and similarities between models. To do so, we extract the MGMs proposed in Sec. 3 for 14 SSL models and use these MGMs to quantify model equivariance-invariance, curvature, and intrinsic dimension for each augmentation manifold.

**Clustering of SSL models:** The similarity between SSL models leads to the clustering illustrated by the dendrogram of Figure 1. In Table 4 (Appendix D) we provide the details of each SSL model and highlight the structural differences that lead to our MGMs-based clustering. To further analyze these clusters, we consider the sparse principal component analysis (PCA) of SSL models having as features the 26 MGMs. We project the MGMs of each SSL model onto the two main components and observe three clusters (see Table 1). We also provide the MGMs that were selected by the sparse PCA and their associated importance in the principal components (see Figure 9 in Appendix D). We note that both simCLR-v1 and simCLR-v2 have a large variation with respect to the features selected by the first principal component. Interestingly, the geometrical property that characterizes both simCLR-v1 and simCLR-v2 is the angle between the semantic and augmented manifold (as indicated by the yellow box (a) in Table 1). This implies that their ability to project the augmented samples onto the data manifold varies significantly with respect to other SSL models. Specifically, while all the models have a low-affinity value between these two directions (orthogonality), both simCLR versions have a high-affinity score ($\approx .8$) meaning that the subspaces spanning the semantic direction and the augmented direction are more aligned. This shows that the dimensional collapse effect observed and analyzed in SimCLR's projection head (19; 21; 20) appears to impact the backbone encoder representation as well. Specifically, simCLR-v1 and simCLR-v2 are the only models projecting the augmentation manifold onto the data manifold such that they are hardly distinguishable. Another distinction of SimCLR models is the low intrinsic dimensionality (see Table 5), again showing that, compared to other SSL models, the SimCLR backbone encoder tends to project the data onto a lower dimensional subspace.

The two other clusters, composed of models having different paradigms are mainly characterized by their differences in terms of semantic equivariance spread. The models in the green cluster (bottom) have low semantic equivariance spread. This implies that InDis, MoCO-v1, PIRL, SeLa-v2, BYOL, DC-v2, SwAV have the same invariance across the input data manifold, i.e., same across all classes. For instance, InDis is highly equivariant to the semantic directions (semantic equivariance = 1.23, as

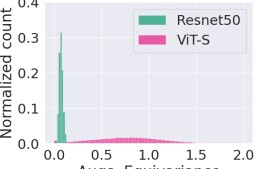 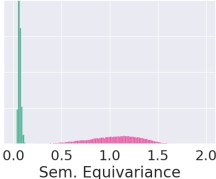 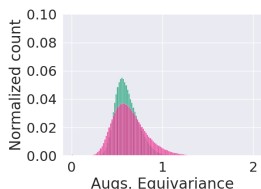 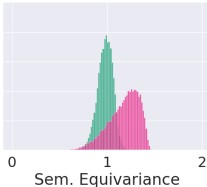

Figure 3: **Histogram of observed equivariance (Semantic, Augmentation)** for convolutional encoder backbone (ResNet50) and a vision transformer backbone (ViT-S) at (**Left**) initialization and (**Right**) after training with same SSL procedure (DINO (32)). We observe that at initialization the inductive bias in convolutional networks leads to an invariant representation with respect to both semantic and augmentation manifold. However, a ViT leads to a more scattered representations of input images belonging to the semantic as well as augmentation manifolds. After training both architectures converge to a similar representation where the marginal variation in spread between the two models impact the performance in downstream tasks.

shown in the $4^{th}$ column), therefore, given its semantic equivariance spread, we know that it is highly equivariant with respect to all ImageNet images.

Another observation from Figure 1 is that when considering *v1* and *v2* versions of PCL, SimCLR, and MoCO, only the two MoCo versions do not belong to the same cluster. The main difference between these two models is their semantic equivariance spread. While MoCo-v1 has low spread in terms of semantic equivariance, MoCo-v2 has a larger spread indicating the equivariance is dependent on the data sample at hand. Note that the main difference between MoCo-v1 and MoCo-v2 is the utilization of a MLP projection head for the latter. Therefore, this observation would suggest that adding a projection head makes the equivariance property of the backbone encoder more dependent on the input data. This result seems intuitive given the fact that the projection head will absorb most of the invariance induced by the SSL loss function.

We also note that some models are strongly invariant to rotations, e.g., SimCLR-v1, SimCLR-v2, while other are strongly equivariant, e.g., PCL-v1, PCL-v2. This is particularly surprising as rotation is typically not part of the augmentation policy used to train SSL models.

**Similarities in SSL models:** From Table 1, we also deduce the geometric properties that do not vary across SSL models. In particular, the semantic and colorjitter affinities, i.e., the curvature of the data manifold along the semantic (label) direction and the colorjitter augmentation direction, show similar behavior across all models studied. This observation shows that the linearization capability of different trained SSL models in these two manifold directions are very similar. We defer further analysis to Appendix C and summarize the observations in Figure 7.

**ViT vs ResNet:** Vision transformers (ViT) (56), built with an architecture inherited from natural language processing (57), have recently emerged as a desirable alternative to convolutional neural networks (CNN) (17) in SSL (32). While ViTs are appealing, as they provide a more general-purpose architecture, the reasons that explain the benefits of the ViT representations over those obtained from CNNs remain unclear. We compare the representations learned by these architectures, focusing on ViT-Small and ResNet50 architectures, as these share similar model capacity (21M vs 23M) and throughput (1007/s vs 1237/s). Figure 3 depicts a geometric comparison of the two architectures using two of our proposed equivariance metrics at initialization as well as after SSL training using DINO (32). We observe that the two architectures lead to very different representations at initialization on both the semantic and augmentation manifold directions. While ResNet50 is biased toward a collapsed representation for all inputs, ViT-S corresponds to a more spread out distribution at initialization. However, after training the two architectures converge to a similar equivariant representation, with ViT-S showing better class separability on the semantic manifold. This observation shows that convolutional networks learn by repulsion where representations are dispersed from a collapsed starting point. On the contrary, ViT shows a different behaviour where it starts from a scattered representation organizing representations by attraction. This could be an explanation for the observed robustness and generality of the representation learned by ViT (58; 32).

## 5 TRANSFER PERFORMANCES OF SSL MODELS

In this section, we investigate the question of determining which geometrical properties are crucial for SSL models to perform better on specific transfer learning tasks. First, we show that the clustering results of SSL models (Sec. 4) based on their geometry mainly coincides with the observed per-cluster transfer performances. We then show that our MGMs can provide intuition regarding the geometric properties that are desirable in order to perform well on specific transfer learning tasks (e.g., classification tasks gain by having rotation invariant representations). Finally, we explore which manifold properties are the most informative for the transfer performance of SSL models in a given task.

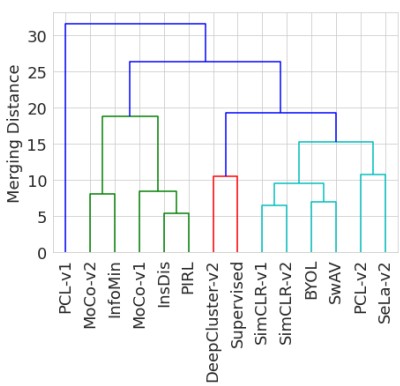

Figure 4: **Dendogram of SSL models based on their transfer learning accuracy.** We observe that the groupings obtained are highly correlated with the clustering performed using our MGMs, thus showing that there is an intrinsic connection between the geometric properties of the SSL model and the transfer learning performances.

**Transfer accuracy based clustering of SSL models:** We depict in Figure 4 the hierarchical clustering of SSL models with respect to their transfer learning accuracy in various tasks. SSL models are clustered if they achieve similar accuracy, good or bad, on a set of tasks. When comparing the clusters obtained using $(i)$ MGMs (displayed in Figure 1), and $(ii)$ transfer learning performances (in Figure 4), we observe that PIRL, InsDis, and MoCo-v1 belong to the same group in both cases. Similarly, SwAV and BYOL, and SimCLR models are highly similar for both clustering results. This confirms our hypothesis that there is a correlation between the geometrical properties of the SSL models and their transfer learning accuracy.

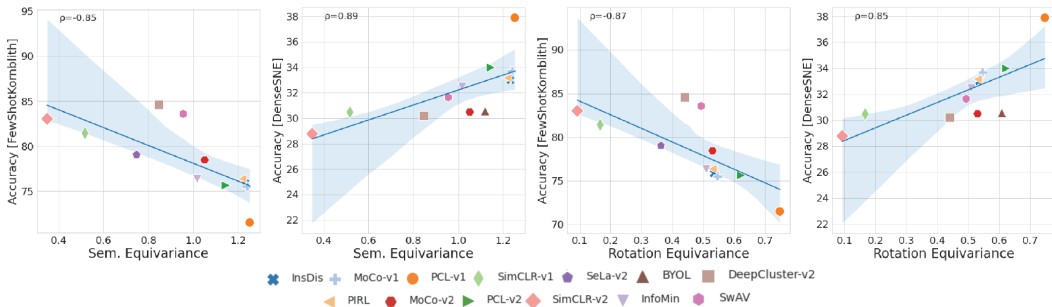

Figure 5: **Correlation between equivariance and transfer learning performances.** We display the pearson coefficient $\rho$ at the top left of each subplot. We observe that the equivariance with respect to semantic direction and rotation of the SSL model is negatively correlated with its capability at performing well in few-shot learning tasks (small domain distance). However, these quantities are positively correlated with the accuracy of the DNN on a dense surface normal estimation task. This observation confirms common intuitions regarding the properties that an embedding should have to transfer accurately on these two tasks. The $p-$values for all results are $\leq 0.01$.

**Recovering geometric properties for transfer learning:**
Intuitively, it is known that to generalize well on image recognition, an SSL model should be invariant to augmentations such as rotation, while to perform well on dense tasks, i.e., pixel-level prediction tasks, it should be equivariant to rotation (26). We confirm this intuition through our proposed MGMs in Figure 5. We show that the semantic and rotation equivariance correlate as expected with the transfer accuracy for two different tasks, namely, *FewShotKornblith* and *DenseSNE*. In particular, we find that for $(i)$ *FewShotKornblith* (few-shot learning with small domain distance), the greater the model invariance to semantic direction and rotation, the better the accuracy, and $(ii)$ *DenseSNE* (dense surface normal estimation) the higher the equivariance of the model the better.

**Exploring geometric properties for transfer learning:** We now propose to explore some possibly counter-intuitive manifold properties of the backbone encoder that correlate with specific transfer learning task. Our method is based on quantifying how well each MGM can explain the per-task transfer capability of SSL models by using simple regression methods capable of performing feature selection. For each task we regress the MGMs onto the transfer learning accuracy. Details and illustrations of the results are shown in Appendix F.

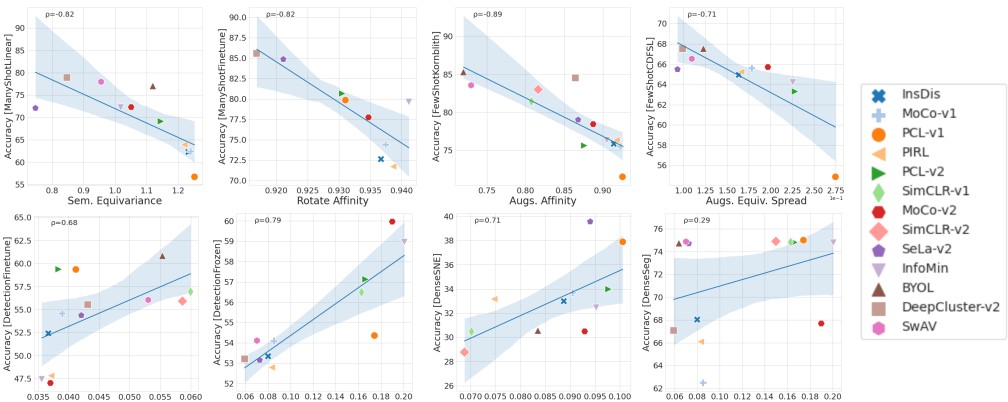

Figure 6: **Correlation between MGMs and transfer performances.** We display the pearson coefficient $\rho$ at the top left of each subplot. We observe that for most transfer learning tasks, there exists a MGM that correlates with the per-task transfer performance. We recover some intuitive results such as for many shot learning linear, higher invariance in the SSL model corresponds to better transfer learning capability. We also note that dense segmentation appears not to be correlated to the considered geometrical metrics of SSL models. The $p-$values for each result was $\leq 0.01$ except for the last plot where the $p$-value$= 0.34$.

We show in Figure 6 the correlation between the selected MGMs and the per-task transfer learning accuracy. We observe that in the case of many shot (linear and fine-tune), few shot (small domain distance and and large domain distance), there exist a negative correlation with intuitive geometric property (Figure 6 *Top*): semantic equivariance, rotation affinity, augmentation affinity, and spread of augmentation equivariance. For the cases of *ManyShotLinear*, *ManyShotFinetune*, and *FewShotKornblith*, we recover geometric properties that were previously considered important for classification (59; 49). However, in the case of *FewShotCDFSL*, where the datasets used differ largely from ImageNet, the transfer performances are correlated with second-order statistics, i.e., the spread, and less on the differences between the average of the MGM distributions. In *DetectionFinetune*, *DetectionFrozen*, *DenseSNE*, and *DenseSeg* tasks, the geometric properties that are correlated with the performances are also second-order statistics (Figure 6 *Bottom*). Specifically, there exists a positive correlation with the spread of crop affinity, semantic equivariance, and colorjitter affinity.

Therefore, we observe that there is an implicit relationship between the geometric properties of SSL models and their transfer learning capabilities. Specifically, for tasks with small transfer distance relative to ImageNet, we observe that the performances of an SSL model are tied to well-known geometric properties, such as invariance and linearization capabilities. While in the case of large transfer distance, we note that the transfer learning capabilities relies on higher-order statistics of the geometry of the backbone encoder, i.e., capturing how the model maps different inputs.

## 6 CONCLUSION

We show that the geometry of SSL models can be efficiently captured by leveraging graph-based metrics. In particular, our data-driven approach provides a way to compare SSL models that can differ in architecture and training paradigms. Our analysis provides insights into the landscape of SSL algorithms, highlighting their geometrical similarities and differences. Further, the proposed geometrical metrics capture the transfer learning capability of SSL models. Our approach motivates the design of transfer learning specific SSL training paradigms that take into account the geometric requirement of the downstream task.

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

## A    ADDITIONAL DETAILS ON EXPERIMENTS

---

**Algorithm 1** MGMs for SSL

---

**Input:**   Dataset $\mathcal{D} = \{x_i\}_{i=1}^{N}$, pre-trained SSL backbone $\boldsymbol{g}$, data augmentation policy $\mathcal{T}$
**Output:**   MGMs for input SSL model and dataset
1: **for** $i = 1$ to $N$ **do**
2:     $\mathcal{E}_i = \{\}$
3:     **for** $t = 1$ to $T$ **do**
4:         Perform random augmentations of $x_i^{(t)} = \mathcal{T}(x_i)$
5:         Extract representation of augmented data $x_i^{(t)}$: $\mathcal{E}_i \cup \{\boldsymbol{g}(x_i^{(t)})\}$
6:     **for** $t = 1$ to $T$ **do**
7:         Compute NNK graph of augmented data in encoder space: $\mathcal{G}_i^{(t)} = \text{NNK}(\boldsymbol{g}(x_i^{(t)}), \mathcal{E}_i)$
8:         Compute local MGM values: $\text{Diam}(\mathcal{G}_i^{(t)})$, $\text{ID}(\mathcal{G}_i^{(t)})$, $\text{Subspace}(\mathcal{G}_i^{(t)})$
9:         Compute curvature MGM $\{ \text{Aff}\big(\text{Subspace}(\mathcal{G}_i^{(t)}), \text{Subspace}(\mathcal{G}_i^{(t')})\big), \quad t = [1, \ldots T], t' \in \mathcal{G}_i^{(t)}\}$

10: Return $\text{MGM}(\boldsymbol{g}, \mathcal{D}, \mathcal{T})$ using local MGM obtained $\{ \text{Diam}_i, \text{ID}_i, \text{Aff}_i, \ i = [1, \ldots N]\}$

---

In Algorithm 1, we present the pseudo-code for geometric evaluation of an SSL model using proposed MGMs as defined in Sec.3, namely, invariance, affinity, and number of neighbors. We first obtain local MGM values computed using the NNK neighborhood $\mathcal{G}_i^{(t)}$ corresponding to each augmented input data forming the graph $\mathcal{G}_i$. A summary of the MGM, its geometric association, and estimation is presented in Table 3. The curvature MGM is evaluated for each augmented data and its neighbor and thus produces one value per edge in the graph while invariance, dimension metrics result in one value per graph constructed. For each metric (MGM), and each direction ($\mathcal{T}$), we obtain a distribution over the entire manifold $\text{MGM}(\boldsymbol{g}, \mathcal{D}, \mathcal{T})$. We summarized this global manifold information by considering the first two moments of the metric distribution (mean and spread) obtained across the dataset. Note that the type of augmentation considered reflects the manifold information corresponding to a specific direction in the feature space, i.e., using rotation augmentation allows us to characterize the embedding manifold induced by the augmentation. For each data, we obtain a metric that provides information regarding the geometric property of the manifold in the considered direction. Cross-augmentation MGMs (e.g. Sem.-Augs. affinity) are obtained by comparing graphs corresponding to different augmentation policies in line 9 of the algorithm, i.e., $\text{Subspace}(\mathcal{G}_i^{(t)})$ obtained with augmentation policy $\mathcal{T}_1$ and $\text{Subspace}(\mathcal{G}_i^{(t')})$ obtained with augmentation policy $\mathcal{T}_2$. Consequently, for evaluating cross-augmentation policies the loop over number of augmentations (lines 3-8) is done twice, one with $\mathcal{T}_1$ and other with $\mathcal{T}_2$.

In our experiments, we evaluate 5 augmentation policies ($\mathcal{T}$) with $T = 50$ augmented samples for each image in the validation set of ImageNet ($N = 50000$ images). This corresponds to constructing 50000 graphs per augmentation policy with each graph containing 50 nodes except for *Semantic* setting where we obtain 1000 graphs with 50 nodes each. Note that each graph construction incorporates 50 NNK neighborhood optimization, one for each node.

## B    DATA AUGMENTATIONS

We follow the data augmentation policy used in (60; 32) which is a typical setting with most self-supervised learning models considered in this work. The augmentations are performed using default settings associated with the augmentation function in PyTorch (61) with the following assigned parameters: random cropping (size=224, interpolation=bicubic), Horizontal flip (p=0.5), colorjitter(brightness=0.4, contrast=0.4, saturation=0.2, hue=0.1) randomly applied with p=0.8, grayscale (p=0.2), Gaussian blur (p=1.0), rotation (degrees=90). The augmentation composition for each setting used in our experiments are presented in Table 2.

| Setting | RandomCrop | HorizontalFlip | Colorjitter | GrayScale | GaussianBlur | Rotation |
|---|---|---|---|---|---|---|
| Sem. Augs. | ✓ | ✓ | ✓ | ✓ | ✓ | |
| Crop | ✓ | | | | | |
| Colorjit. | | | ✓ | | | |
| Rotate | | | | | | ✓ |

Table 2: Evaluation setting and composed augmentation functions (sequentially applied in the order listed from left to right). All images are resized to 224 (bicubic interpolation) if not randomly cropped to the size and are mean-centered and standardized along each color channel based on statistics obtained with the ImageNet training set.

| Metric | Geometric property | Estimation |
|---|---|---|
| Polytope diameter | Invariance/Equivariance | The maximum distance between the neighbors of an NNK polytope $\max_{k,l \in \mathcal{N}(\boldsymbol{f}(\boldsymbol{x}_i)))} \left\| \hat{\boldsymbol{f}}(\boldsymbol{x}_k) - \hat{\boldsymbol{f}}(\boldsymbol{x}_l)) \right\|_2$ |
| Affinity | Local curvature | The cosine similarity of principal components of two neighboring NNK polytopes $\sqrt{\frac{\cos^2(\theta_1) + \cdots + \cos^2(\theta_{n_i n_j})}{n_i n_j}}$ |
| No. of neighbors | Intrinsic dimension | The nonzero weighted neighbors identified by NNK $\mathrm{Card}(\mathcal{N}(\boldsymbol{f}(\boldsymbol{x}_i)))$ |

Table 3: Proposed Manifold Graph Metrics, their relationship to the geometric property of the manifold, and method of estimation using observed embeddings. Note that a similar diameter evaluation with a $k$-nearest neighbor will explicitly depend on the choice of $k$ while using 1-nearest neighbor will reduce the local geometry to only one direction. Thus, the use of a neighborhood definition that is adaptive to the local geometry of the data is crucial to successfully observe the properties of the manifold.

# C    SECTION 3 - ADDITIONAL FIGURES

**Interpreting the MGM mean and spread:**    The local MGM values are calculated from a distribution over the entire dataset. As noted earlier, we compute the mean and the standard deviation (referred to as the spread) of this distribution to capture the characteristics of an SSL model. The interpretation of the mean is straightforward and can be seen as the (average) equivariance, curvature, or dimension of the augmentation manifold embedded using the SSL model on a dataset. The spread of the metric captures variation of the geometric property with respect to differences in the input image. For example, an SSL model can be equally invariant to all input images in the dataset (equivariance spread small) while another model might show better invariance with respect to some classes while not with some other classes (equivariance spread large) but have the same average equivariance as the first model across the dataset. In this work, we do not perform per-class based analysis since the interest was to study the impact of global characteristics with that of downstream performance which are in general coarse dataset-level measures. However, we believe that the MGMs proposed can be extended to specific class/attribute-level analysis. Furthermore, we expect such studies can help capture representational heteroscedasticity in an SSL model or how the model adapts to domain shift during transfer (e.g., sketch images to real images of an object), to name a few.

**Variability in SSL models:**    In Figure 7, for each of the MGMs studied, we depict the observed variability in these metrics across all models. The affinity between the semantic manifold and augmentations manifold (Sem.-Augs.) is the MGM capturing the main differences across models. This metric can be intuitively understood as the angle between the natural image manifold and the manifold produced by the data augmentation policy. This implies that the ability of an SSL model to project the augmented samples onto the data manifold varies significantly. The second important feature capturing the dissimilarity between SSL models is the equivariance spread along semantic and colorjitter directions. It implies that while some SSL models have a lot of variation in their invariance, others show equal invariance across the entire manifold. Finally, another geometrical feature that highly contributes to the dissimilarity between SSL models is the equivariance to rotation. Interestingly, some models will be strongly invariant to rotations while other will be strongly equivariant. Note that rotation is not part of the augmentation policy of the models we evaluate.

We also visualize in Figure 7 the geometrical properties that do not vary across SSL models. The main ones are the semantic and colorjitter affinities, i.e., the curvature of the data manifold along the semantic (label) direction as well as the colorjitter augmentation direction. This observation shows that the linearization capability of SSL models regarding these two manifold directions is highly similar.

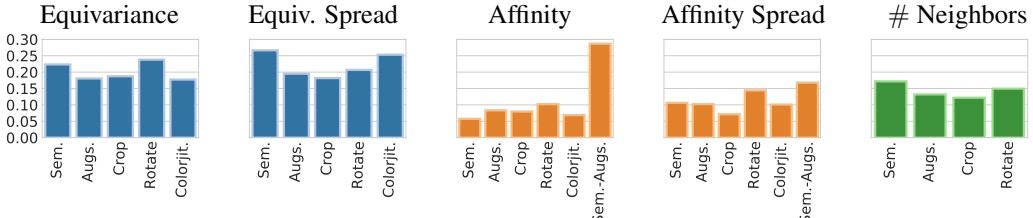

Figure 7: **Manifold graph metrics variability across SSL models.** We depict the variation in evaluated MGMs across 14 pretrained SSL models considered in this paper. We note that the differences in the the different SSL model embedding can be summarized using five manifold properties, namely, ($i$) Sem. equivariance, ($ii$) Rotate equivariance, ($iii$) Sem. equivariance spread, ($iv$) Colorjit. equivariance spread, ($v$) Sem.-Aug. affinity. These properties highlight the key manifold properties with largest variation across the different SSL models.

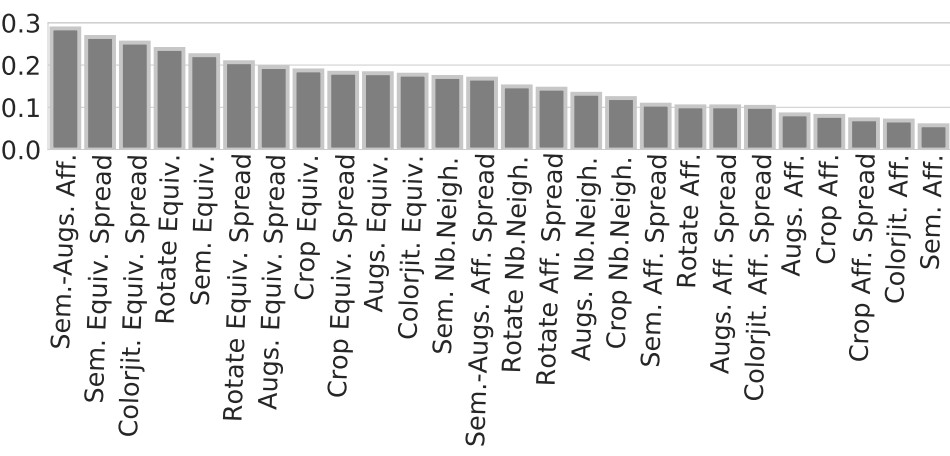

Figure 8: **(Sorted) Evaluation of manifold graph metrics variability across SSL model.** We compute here the standard deviation of normalized manifold graph metrics to highlight the varying manifold properties across SSL models. We observe that five manifold properties explain most of the differences in SSL models: $(i)$ affinity between semantic and augmentations manifold , $(ii)$ spread of semantic equivariance, $(iii)$spread of colorjitter equivariance , $(iv)$ rotation equivariance, $(v)$ semantic equivariance

# D  SECTION 4 - ADDITIONAL FIGURES

## D.1  CLUSTERING ALGORITHM DETAILS

We used the python Scipy library's linkage module to performing hierarchical agglomerative clustering. The method we used to compute the distance is based on the *Voor Hees Algorithm*. In Figure 4, we use the transfer learning accuracy of each model with respect to all the tasks defined in the experimental settings in section 3. That is, the features of the clustering algorithms are the per-model accuracy on the task. The per-task performances are not mentioned here as it was already developed in referenced paper (12). We propose this result to display the similarity between the clusters based on transfer learning performances and the clusters based on the manifold graph metrics.

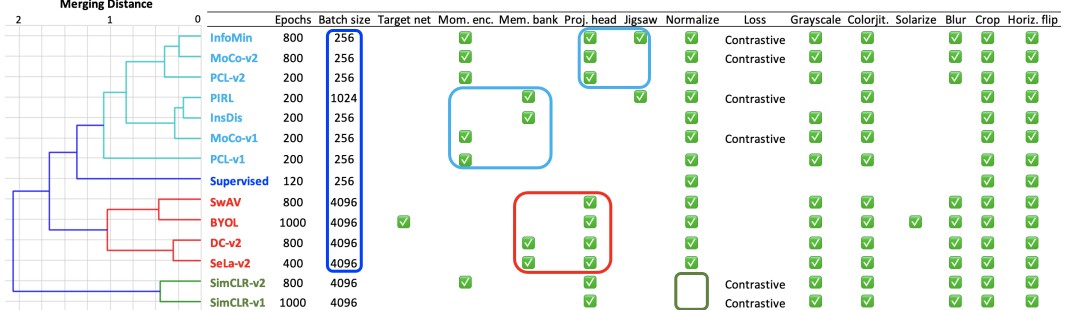

Table 4: **SSL model details and the associated dendrogram split learned by our geometric metrics.** We highlight the structural differences between SSL models that would possibly correspond to the geometric differences and similarities we observe. It appears that the main difference between SimCLR models and all the other SSL models could be explained by input normalization. Then, the batch size appears to also affect drastically the geometry. Finally, we observe on a finer scale that momentum encoder, the presence of projection head and memory bank could also lead to geometric differences. We believe that the current classification of SSL approaches (contrastive, non-contrastive, cluster-based) is not sufficient to capture their geometric differences. All the parameters described in here should be taken into account as we believe there exists a complex interplay between these choices and the induced geometry of the trained SSL model.

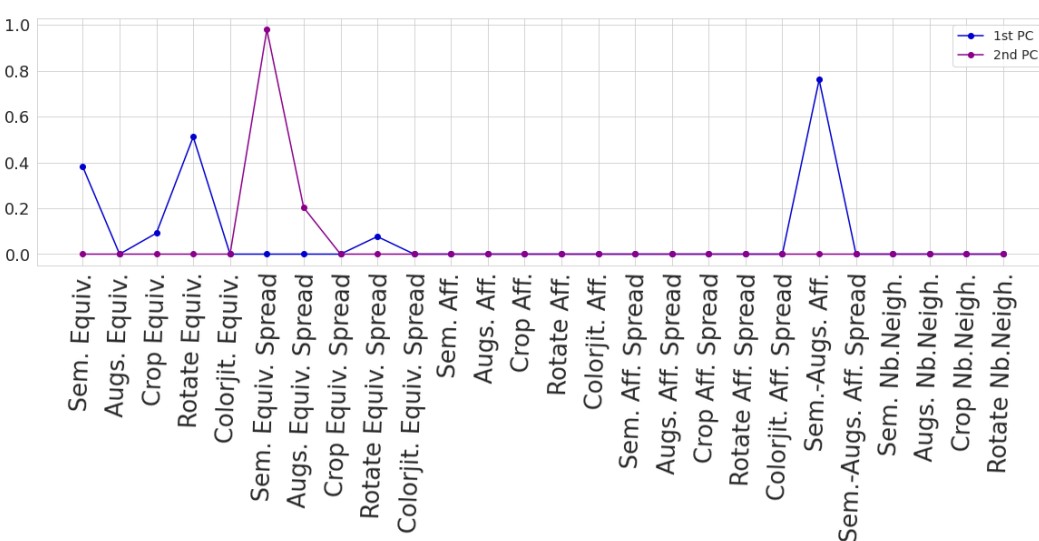

Figure 9: **Absolute value of principal components of per-feature manifold graph metrics matrix.** We display here the absolute value of the two principal components obtained after sparse PCA of the matrix of dimension number of models × number of graph manifold metrics. This corresponds to the principal components used to visualize the distribution of the SSL models in the two-dimensional plane as in Figure 1. We observe that few manifold graph metrics encapsulate most of the variance contained across SSL models; namely: Semantic Equivariance, Augmentations Equivariance, Crop Equivariance, Rotation Equivariance, Semantic Equivariance Spread, Augmentations Equivariance Spread, Rotation Equivariance Spread, Colorjitter Equivariance Spread, Semantic-Augmentations Affinity.

# E    SECTION 5 - ADDITIONAL FIGURES

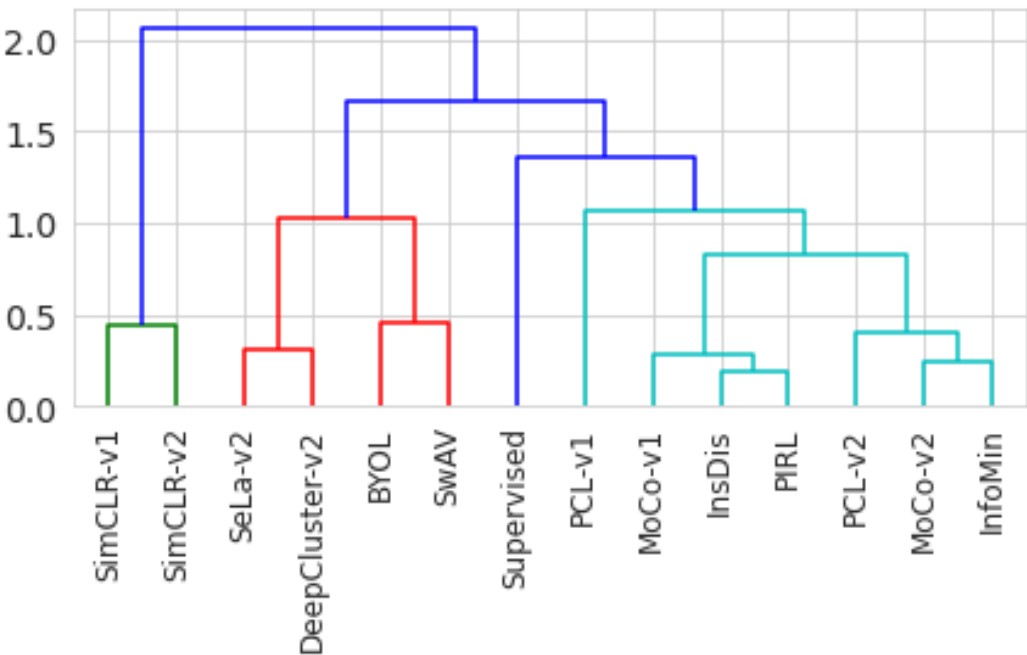

Figure 10: **Dendrogram of SSL models**. We compute the dendrogram of the MGM of SSL models. This shows us that there are in fact three main classes of SSL models in terms of manifold property as well as the proximity between different models. This result also confirms the clustering based on PCA visualized in Figure 1. In particular, SimCLR-v1 and SimCLR-v2 appear to be the most distant model from all the SSL paradigms tested here. Interestingly, the clustering obtained here does not correspond to the different classes of SSL algorithms: contrastive, non-contrastive, cluster-based, and memory-based.

# F    FEATURE SELECTION

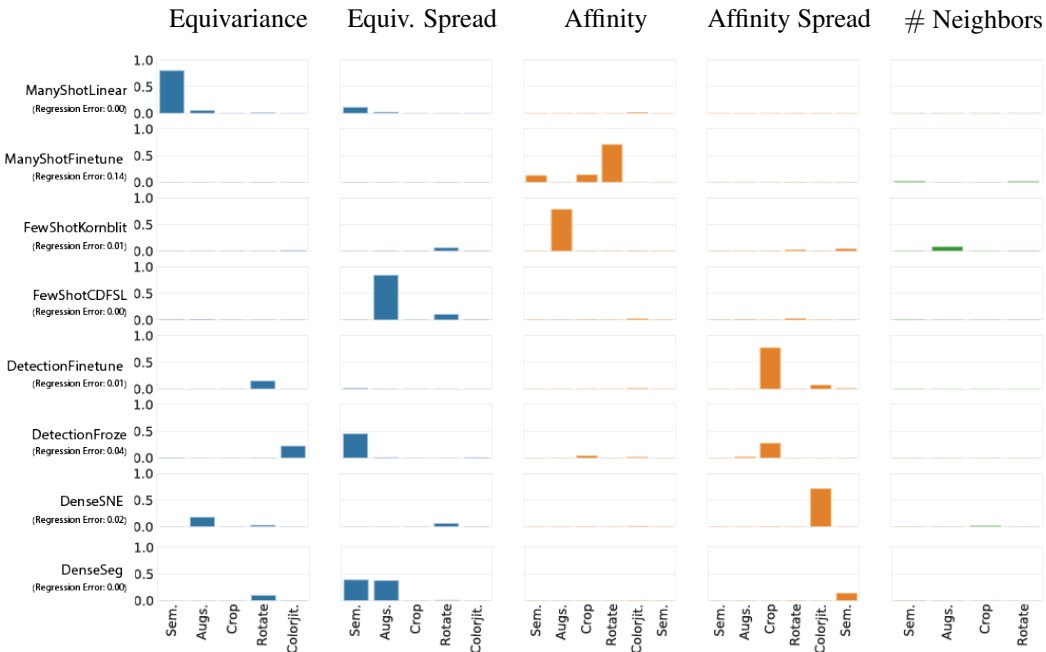

Figure 11: **Per transfer learning task MGM importance - Decision Tree.** For each transfer learning task, we exploit the feature selection of decision trees (depth= 5) to visualize which MGM are crucial to characterize the transfer learning accuracy. To do so, we fit the MGMs with the transfer learning accuracies. In this figure, we highlight the MGMs that explain the most the per-task transfer learning accuracy. Note that we are not interested by the regression error, but by the importance of each MGM for predicting each transfer learning task. While being intuitive, this result shows that mainly the invariance and curvature of the DNN characterize their transfer learning capability. The first observation is that the intrinsic dimension of the DNN (displayed in green as # Neighbors) does not allow one to characterize any task-specific transfer learning accuracy. Depending on the task, different geometrical properties matter. For many shot linear, the equivariance to semantic direction of the data manifold is crucial. For few shot with small transfer domain distance, the linearization capability of the DNN with respect to the augmentations captures most of the information regarding the transfer learning capability. While most tasks appear to be explained by a single of few MGMs, in the case of frozen detection and dense segmentation, multiple MGMs are required to explain the transfer learning capability of SSL models.

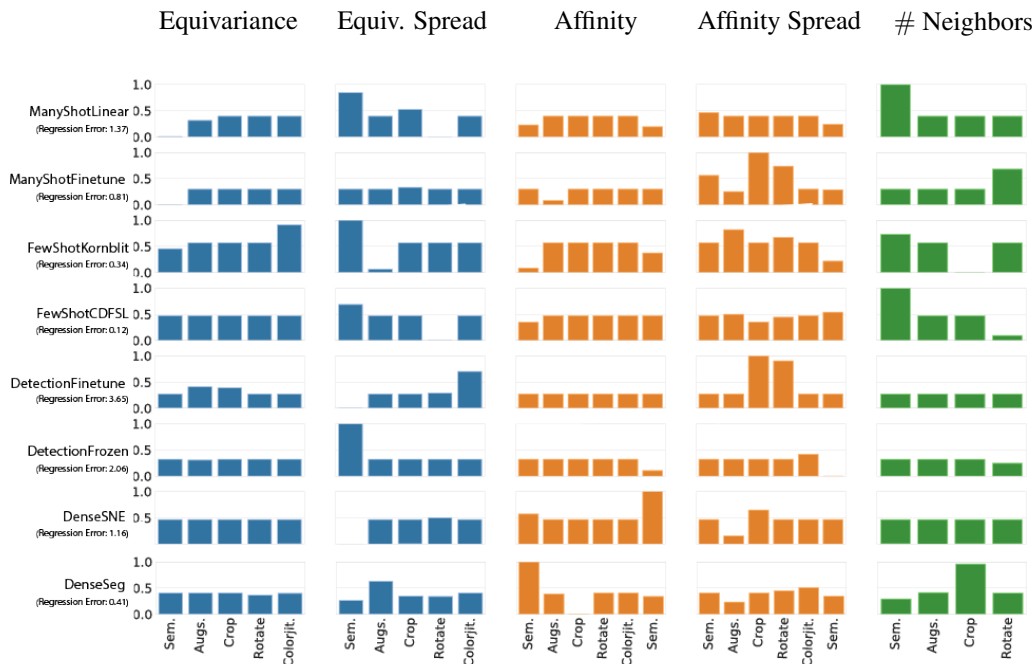

Figure 12: **MGM Importance Per Transfer Learning Task - Lasso.** For each transfer learning task, we exploit the feature selection of LASSO method to visualize which MGM are crucial to characterize the transfer learning accuracy. As intuitively expected, we observe that depending of the transfer learning task specific manifold properties are more important. For instance, for dense detection, the equivariance spread of the semantic manifold highly correlates with the transfer learning accuracy. Indicating that the variation of equivariance/invariance induced by the DNN manifold w.r.t. the input data is critical to accomplish dense detection. Similarly, for dense segmentation, the curvature of the semantic manifold appears to be the most important feature.

# G MGMs Values for Each SSL Model

| Model | Equivariance | | | | | Equivariance Spread | | | | | Affinity | | | | | | Affinity Spread | | | | | | No. of neighbors | | | |
|---|---|---|---|---|---|---|---|---|---|---|---|---|---|---|---|---|---|---|---|---|---|---|---|---|---|---|
| | Sem. | Augs. | Crop | Rotate | Colorjit. | Sem. | Augs. | Crop | Rotate | Colorjit. | Sem. | Augs. | Crop | Rotate | Colorjit. | Sem.-Augs. | Sem. | Augs. | Crop | Rotate | Colorjit. | Sem.-Augs. | Sem. | Augs. | Crop | Rotate |
| InsDis | 1.23 | 0.77 | 0.45 | 0.53 | 0.12 | 0.08 | 0.16 | 0.14 | 0.16 | 0.06 | 0.80 | 0.91 | 0.89 | 0.94 | 0.72 | 0.22 | 0.05 | 0.04 | 0.05 | 0.04 | 0.09 | 0.05 | 10.48 | 6.89 | 7.28 | 5.69 |
| MoCo-v1 | 1.24 | 0.76 | 0.44 | 0.55 | 0.11 | 0.09 | 0.18 | 0.16 | 0.18 | 0.05 | 0.81 | 0.92 | 0.90 | 0.94 | 0.73 | 0.21 | 0.05 | 0.04 | 0.05 | 0.04 | 0.09 | 0.06 | 9.95 | 6.57 | 6.79 | 5.18 |
| PIRL | 1.22 | 0.77 | 0.45 | 0.53 | 0.14 | 0.08 | 0.17 | 0.16 | 0.17 | 0.06 | 0.80 | 0.92 | 0.90 | 0.94 | 0.76 | 0.22 | 0.05 | 0.04 | 0.05 | 0.04 | 0.07 | 0.05 | 10.20 | 6.83 | 7.30 | 5.51 |
| SeLa-v2 | 0.75 | 0.44 | 0.38 | 0.36 | 0.09 | 0.07 | 0.09 | 0.11 | 0.09 | 0.04 | 0.82 | 0.87 | 0.89 | 0.92 | 0.70 | 0.14 | 0.06 | 0.05 | 0.05 | 0.04 | 0.09 | 0.03 | 10.42 | 8.28 | 7.77 | 6.54 |
| BYOL | 1.12 | 0.80 | 0.65 | 0.61 | 0.16 | 0.06 | 0.12 | 0.16 | 0.13 | 0.07 | 0.71 | 0.72 | 0.73 | 0.73 | 0.74 | 0.18 | 0.04 | 0.04 | 0.05 | 0.06 | 0.08 | 0.03 | 12.46 | 8.73 | 7.95 | 6.48 |
| DC-v2 | 0.85 | 0.54 | 0.45 | 0.44 | 0.10 | 0.06 | 0.10 | 0.12 | 0.10 | 0.04 | 0.79 | 0.86 | 0.88 | 0.92 | 0.70 | 0.16 | 0.05 | 0.05 | 0.06 | 0.04 | 0.09 | 0.03 | 11.80 | 8.36 | 7.94 | 6.57 |
| SwAV | 0.96 | 0.60 | 0.50 | 0.49 | 0.11 | 0.07 | 0.11 | 0.13 | 0.11 | 0.05 | 0.71 | 0.73 | 0.74 | 0.74 | 0.72 | 0.17 | 0.04 | 0.04 | 0.05 | 0.05 | 0.08 | 0.03 | 11.49 | 8.39 | 8.00 | 6.70 |
| SimCLR-v1 | 0.52 | 0.52 | 0.29 | 0.17 | 0.15 | 0.16 | 0.20 | 0.14 | 0.07 | 0.15 | 0.81 | 0.81 | 0.79 | 0.75 | 0.89 | 0.81 | 0.04 | 0.04 | 0.05 | 0.06 | 0.07 | 0.05 | 5.52 | 5.40 | 5.26 | 3.81 |
| SimCLR-v2 | 0.35 | 0.32 | 0.19 | 0.09 | 0.11 | 0.15 | 0.20 | 0.12 | 0.06 | 0.14 | 0.82 | 0.82 | 0.80 | 0.76 | 0.90 | 0.82 | 0.04 | 0.04 | 0.05 | 0.06 | 0.07 | 0.04 | 5.29 | 4.91 | 4.98 | 3.33 |
| PCL-v1 | 1.25 | 0.82 | 0.51 | 0.75 | 0.11 | 0.17 | 0.27 | 0.28 | 0.29 | 0.08 | 0.87 | 0.93 | 0.92 | 0.93 | 0.76 | 0.10 | 0.05 | 0.04 | 0.04 | 0.04 | 0.10 | 0.05 | 7.38 | 6.40 | 5.91 | 5.09 |
| PCL-v2 | 1.15 | 0.65 | 0.57 | 0.62 | 0.12 | 0.17 | 0.23 | 0.23 | 0.19 | 0.07 | 0.83 | 0.88 | 0.90 | 0.93 | 0.72 | 0.17 | 0.05 | 0.05 | 0.05 | 0.04 | 0.10 | 0.04 | 9.67 | 7.94 | 7.25 | 5.99 |
| MoCo-v2 | 1.05 | 0.55 | 0.46 | 0.53 | 0.10 | 0.19 | 0.20 | 0.20 | 0.17 | 0.06 | 0.83 | 0.89 | 0.91 | 0.93 | 0.73 | 0.19 | 0.05 | 0.05 | 0.05 | 0.04 | 0.09 | 0.04 | 9.49 | 7.52 | 6.96 | 5.80 |
| InfoMin | 1.02 | 0.58 | 0.45 | 0.51 | 0.11 | 0.20 | 0.23 | 0.23 | 0.17 | 0.07 | 0.84 | 0.91 | 0.92 | 0.94 | 0.75 | 0.20 | 0.05 | 0.04 | 0.04 | 0.04 | 0.10 | 0.04 | 8.81 | 7.27 | 6.75 | 5.63 |
| Supervised | 1.10 | 0.94 | 0.68 | 0.64 | 0.31 | 0.17 | 0.15 | 0.19 | 0.14 | 0.15 | 0.72 | 0.72 | 0.73 | 0.72 | 0.75 | 0.14 | 0.06 | 0.04 | 0.05 | 0.05 | 0.08 | 0.04 | 10.18 | 8.00 | 7.60 | 7.42 |

Table 5: **Proposed MGMs observed for different SSL models.** We display here the values of each MGM for each SSL model. These correspond to the 26 MGMs that we consider in all the analysis we provide in the paper.

# H Simulated Experiments

In this section, we present a study on controlled manifold settings to demonstrate the ability of the proposed metric in capturing the geometry of the manifold. Using data points ($N = 1024$) sampled from three 10-dimensional manifolds embedded in an 128-dimensional feature space (Figure 13). Further, we generate images corresponding to these embeddings using an ImageNet pretrained generative model BigGAN as in (62) to show the implication of the geometry and its relationship with respect to images. We restrict the generation to a single ImageNet class, namely, *basenji*. The images generated are of size $128 \times 128 \times 3$ This setup is equivalent to knowing the augmentation manifold and then observing the images corresponding to this manifold.

- Random samples: Here, we sample 1024 embeddings from a 10-dimensional multivariate Gaussian to simulate an embedding space where no explicit geometry exists.
- Linear subspace with uniformly spaced samples: We start with 8 seed embeddings (equivalent to unique image inputs) and then 128 embeddings (equivalent to augmentations of the images) obtained by uniformly spaced translation of the seed embeddings.
- Spherical linear interpolated (SLERP) samples: In this setup, we start with *pairs* of 8 seed embeddings and then 128 augmentation embeddings that are obtained via a symmetric weighted sum of each pair of endpoints.

Note that, the latter two settings correspond to a setting where the local polytopes corresponding to neighbors will have high affinity owing to the smooth curvature and equal invariance (diameter) by design. In contrast, the geometry of the randomly sampled data will have low affinity and high diameter. This is because the neighboring polytopes will not be aligned with neighbors within each polytope possibly spanning the entire subspace[1].

The image generation using BigGAN presents a unique opportunity to test and observe the relationship between the embedding manifold and the images. Of course, typically one is interested in the reverse direction where one has input images and would like to observe the embeddings (as in our SSL study). But this simulation allows us to explicitly study the relationship between different manifolds and their MGM metrics relative to the image mapping of the embeddings.

# I Transfer Learning Experiments Detail

## I.1 Many Shot Datasets

FGVC Aircraft (63) , Caltech-101 (64), Stanford Cars (65), CIFAR10 (66), DTD (67), Oxford 102 Flowers (68), Food-101 (69), Oxford-IIIT Pets (70), SUN397 (71) and Pascal VOC2007 (53).

## I.2 Few Shot Datasets

Few Shot-Kornblith corresponds to the same datasets used for many shot learning except for VOC2007. For CDFSL, we use the Few-Shot Learning benchmark introduced by (52). It consists of 4 datasets that exhibit

---

[1]In the simulated experiment we present the angle between the polytopes instead of the affinity. These two metrics are directly related to each other in that one is the cosine of the other. This choice makes the local curvature property explicit for readers not familiar with affinity used in subspace literature.

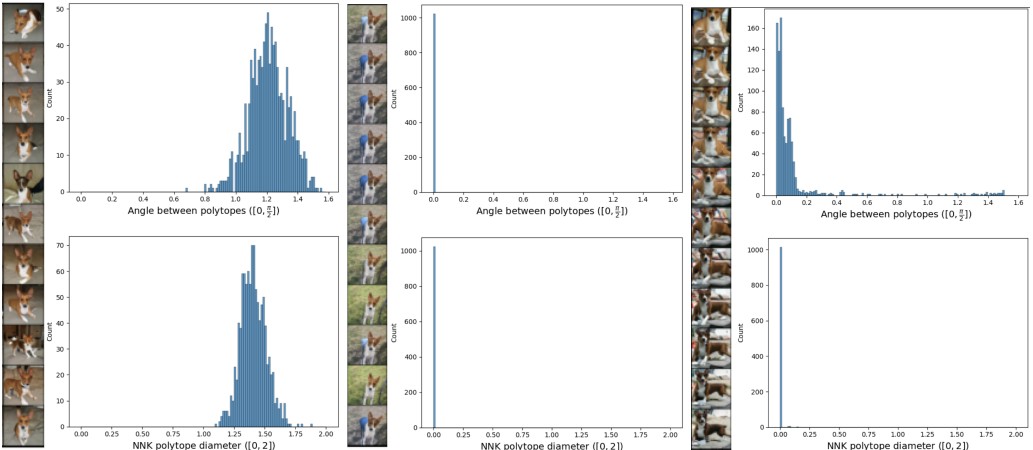

Figure 13: Sample BigGAN images, Observed NNK diameter (Invariance), and Angle between neighboring NNK polytopes (Affinity/Curvature) for three embedding manifold settings: Random (**Left**), Linear (**Middle**), and Spherical (**Right**). As expected, the randomly sampled examples produce neighboring polytopes that are oriented almost orthogonal with respect to each other indicative of the absence of a smooth manifold. Further, the large diameter in this setting shows that the examples are locally distinct i.e., there is no collapse in the representation of the neighbors that form the polytope. In contrast, we see that the embeddings from linear and spherical subspaces have polytopes that are in close relationship to each other and are locally invariant.

increasing dissimilarity to natural images, CropDiseases (72), EuroSAT (73), ISIC2018 (74; 75), and ChestX (76).

### I.3 DETECTION

For detection, we use PASCAL VOC dataset (53).

## J LIMITATIONS

In this work, we performed our geometric analysis using ImageNet validation dataset on SSL models with a ResNet50 backbone and one ViT architecture with approximately same number of parameters. Our results are based on the feature space embedding of the models and we do not foresee any issues when scaling up to larger models and datasets, or even different modalities, but this remains to be tested and is left open for future work. We also restricted our analysis to 5 augmentation settings, which we considered to be of practical relevance, but further exploration is required to better understand the properties of the embedding manifold. Although we show empirically the correlation between geometrical properties and several downstream tasks, we did not explore the theoretical implication of having a certain geometry and its relationship to transfer generalization. We emphasize that our formulation, unlike the accuracy metric previously studied (12; 25), is amenable to theoretical study using spectral and graph theoretical concepts. We note that our work can be leveraged in conjunction with previously developed approaches, such as (77; 78), to induce a particular embedding geometry depending on the transfer learning application at hand. Our focus in this work was to provide a big picture tool for understanding features obtained with SSL models using geometry/graphs and hope that it leads to new ideas for understanding and training deep learning models.

