# OpenReview forum: "The Geometry of Self-supervised Learning Models and its Impact on Transfer Learning"
_ICLR.cc/2023/Conference — Submitted to ICLR 2023_

### Official Review · Reviewer_HJu3 · 2022-10-23

**Confidence:** 3
**Correctness:** 2
**Technical Novelty And Significance:** 2
**Empirical Novelty And Significance:** 2
**Recommendation:** 3

**Clarity, Quality, Novelty And Reproducibility:**

The paper is well written and easy to follow, though some of the technical details are hidden and organization could be improved. As a reviewer, I can understand the methods for evaluation and the intuition behind their metrics. To improve the organization a table of the MGM descriptions, equations, and method of calculation would better organize the paper and make for easier digestion. This paper has a lot of evaluation settings as well, and the authors should consider whether organizing these into a table for ease of understanding would be beneficial as well. These tables could be placed in the appendix.

The value of the metrics is not entirely obvious, beyond the analysis provided in the paper. The conclusion that their metrics motivate the design of new SSL approaches begs the question: why did they not test this conclusion? The paper offers no concrete evidence that these metrics can be used directly for self-supervision. Furthermore, it is unclear to me why the authors limited the application to self-supervised learning. Why would these geometric features not be useful for other models? E.g. as regularization for semi-supervised models or even supervised models.

The metrics introduced seem to be novel. However, as I mentioned above, the degree to which they express the qualitative features indicated by their names in the networks is not clear. The evaluation work is certainly novel, however I again question the value of this analysis. Clustering networks based on these metrics involves a lot of simplification. For instance, why should the properties studied here by isotropic? They could certainly vary throughout the input space. This seems to limit the value of the uniform analysis applied for the entire network as if the metrics were not spatially varying in the input space(since the averages and spread are computed over a large collection of samples). The implication is that these are uniform behaviors across input space, but this is clearly not the case (imagine images from different domains with varying complexity).

The authors did not include a reproducibility statement in their work, however, assuming they used open source models/training code for the SSL methods here reproducing their work should be possible by a motivated scientist.

**Strength And Weaknesses:**

The main strength of the work is that the objective of the paper seems novel and is clearly intelligible.

There are several weaknesses in my opinion. First of all, these choices of metrics are arbitrary: while the authors claim the diameter and affinity metrics have semantic meanings in terms of equivariant and intrinsic dimension, this is not corroborated by evidence. The minimal evidence provided is in the terms of post-hoc reasoning about the nature of the models on some datasets, in light of their MGM values. This does not hold up under scrutiny, since e.g. the effectiveness of the diameter metric at capturing equivariance will vary from model to model and from training strategy to training strategy. This is because these networks have high dimensional encodings and so the encodings can be far apart geometrically, but look similar to the self-supervised decision heads (e.g. the momentum encoder weights or the prototype weights, etc.).  There would need to be more analysis specifically on the MGMs looking at the uniformity of these metrics across a range of networks and training regimes to a establish the significance of them in the proposed analysis. Are the MGMs robust to hyperparameters or small changes in the SSL methods? How about other deep learning representations? There is not enough analysis dedicated to the value of metrics independent of the clustering and correlation analyses and that leads to the another weaknness.
All of the statements in this work rely on some kind of dimensionality reduction or summarization of the aggregated statistics of the MGMs across a large input set (Imagenet validation). For me, it is hard to take this kind of analysis seriously given that you are reducing very large and complex collections of weights and data (generally hundred of megabytes of networks, plus gigabytes of images) into dendrograms and x-y plots. Given the lack of analysis on the MGMs themselves, I do not feel I have any intuition on the significance of this kind of analysis or can infer whether it is "right". Slink is a nonparametric clustering approach, but the linkages are across different algorithms represented by MGMs. Again, this leaves the reader wondering what the significance of the linkages is---beyond a degree (distance in the MGM space) or a linkage (the closest neighbor under the slink algorithm used here).

**Summary Of The Paper:**

This work attempts to establish general geometric statistics of deep learning representations (Manifold graph metrics---MGMs) that can be useful for quantifying various properties of deep learning models. According to the paper, these can be viewed as measurements of desirable qualitative features of the resulting models. The paper makes statements about the relationships between the self-supervised models (SSL) based on the statistics developed. Specifically, the authors claim that SSL models can be clustered usefully by these geometric properties, and that they are a strong indicator of transfer learning capabilities. They do so by analyzing clustering of the SSL models on both the MGM space as well as the accuracy space (across a range of tasks).

To create a set of clusterable features, the authors consider the space of MGMs applied to manifolds resulting from neighborhoods derived from 5 generating operations: semantic category, augmentations, crop, colorjitter, and rotations. For each of these they compute the mean and spread of: equivariance-invariance (diameter of the polytope resulting from NNK performed on the set generated by the mechanism) and affinity (the angle between the natural and augmented manifolds resulting from the NNK procedure) as well as the cardinality of the NNK neighborhood (which is a surrogate for dimension of the underlying manifold).

From these sets of features cluster and correlation analysis is applied. The authors compare the clusters resulting from features comprised by accuracy across a variety of tasks to that of the MGM features. They claim that this shows that the there is an intrinsic connection between the geometric properties of the SSL model (MGMs in this context) and the transfer learning performances. The authors close with a claim that the MGMs could be used for improved SSL.

**Summary Of The Review:**

This work proposes some novel descriptor of deep learning representations viewed through augmented inference. The authors then apply these descriptors to understanding the geometry of self-supervised deep learning representations as applied to computer vision. This aspect of the work is novel. The authors do not provide in depth analysis of their proposed descriptors, convincing the reader of their value and stated significance. By directly applying correlation and clustering analysis to these descriptors without providing compelling evidence or intuition of the meaning, the authors leave the reader uncertain of the actual result here. More in depth understanding of their proposed descriptors and whether they can actually provide the evidence of the claims made in the paper should be performed. Finally, the authors allude to the value of these descriptors for self-supervised learning. But they do not conduct any experiments towards this end. It is worth noting that conducting such experiments would go a ways towards resolving the complaint of the lack of evidence of the value of the descriptors noted earlier.

---

> ### Author Response · Authors · 2022-11-19
> **Response to Reviewer HJu3 - MGM validation and clarification**
>
> We thank the reviewer for the feedback. The reviewer acknowledges the novelty of the proposed metrics but expresses serious concerns about whether these metrics can be trusted.  The fundamental concern raised by the reviewer is that we have not proven that our proposed metrics (e.g., diameter) can be compared across different flavors of SSL training and the only justification of their validity is post hoc, i.e., based on correlation to performance on transfer tasks. In what follows we provide a more detailed justification of why these metrics, and the corresponding comparisons,  are meaningful. We also answer other concerns raised by the reviewer.
>
> `
> First of all, these choices of metrics are arbitrary: while the authors claim the diameter and affinity metrics have semantic meanings in terms of equivariant and intrinsic dimension, this is not corroborated by evidence.  This is because these networks have high dimensional encodings and so the encodings can be far apart geometrically, but look similar to the self-supervised decision heads (e.g. the momentum encoder weights or the prototype weights, etc.). There would need to be more analysis specifically on the MGMs looking at the uniformity of these metrics across a range of networks and training regimes to establish the significance of them in the proposed analysis.`
>
> `However, as I mentioned above, the degree to which they express the qualitative features indicated by their names in the networks is not clear. The evaluation work is certainly novel, however I again question the value of this analysis.
> `
> - The reviewer raises concern that the properties of the embeddings (e.g., scale) might be different across the SSL networks and thus the distances (e.g., diameters) computed for each SSL network might not be comparable.
>  While this was perhaps not sufficiently emphasized in the original version of the paper, our system is in fact designed to make the results comparable across SSL flavors. Specifically:
> 1) The different SSL models that are compared in our work all share the same backbone architecture (ResNet50) and thus embed images in a space having the same dimension;
> 2)  In order  for the computed diameters to  be comparable  across different networks we normalize the embeddings so that the  distances between the embeddings of two different images are always in the range 0 to 2;
> 3) The embedding from ResNet50 SSL encoder is passed through batch normalization layers (mean-centered and variance rescaled to 1) and justifies further the embedding comparison across the SSL models.
>
> Given that distances are normalized the diameters are not based on absolute distances, which could be different for different SSL approaches, instead are normalized with respect to a common range of distances across SSL methods  Also, note that the affinity metric, used as an alternative to diameter, is related to the direction and is less dependent on the scale of the distances.
>
> The reviewer might be right to question the result if the comparison was across different SSL backbones and if the embedding spaces had vastly different dimensions but care was taken to ensure that these models are compared fairly.
> - The reviewer correctly points out that the self-supervised decision~(projector) head can adjust to the degree of invariance in the encoder space. However, we note that in practice for transfer one typically works with the encoder space representations and ignores the projector head. Thus,  to understand transfer performance it is more important to study the embeddings in the encoder space (as we propose) and ignore how these are modified by the projector.
>  - To further establish the validity of our proposed metrics, we have updated our paper with additional experiments on simulated manifolds/embeddings to show that the metrics do capture geometric properties - Appendix section H, Figure 13.
> - Note that the importance of invariance has been one of the main arguments behind the development of self-supervised learning. Yet, up to this point, there has been no efficient way to capture and analyze this property from a geometrical standpoint. Our work proposes such an approach, which we believe contributes to a much better understanding of the role of invariance in deep networks.

---

> ### Author Response · Authors · 2022-11-19
> **Response to Reviewer HJu3 - Robustness and analysis clarification**
>
> `Are the MGMs robust to hyperparameters or small changes in the SSL methods? How about other deep learning representations? There is not enough analysis dedicated to the value of metrics independent of the clustering and correlation analyses and that leads to the another weakness. `
>
> - We note that the geometry is associated in complex ways with the training setup. The analysis is for given a model, what is the geometry? This does not imply that if one changes the training setup slightly the model will have the same geometry - In fact, our analysis shows that the geometry is affected by simple changes such as batch normalization and batch size.
> - We do not claim that if one trains with say Moco-v2 loss, one will have a particular geometry. The claim is that given a trained model one can evaluate geometric metrics on this model (similar to evaluating the accuracy of the model on ImageNet) and that these metrics have a direct relationship to transfer performance.
>
> `All of the statements in this work rely on some kind of dimensionality reduction or summarization of the aggregated statistics of the MGMs across a large input set (Imagenet validation). For me, it is hard to take this kind of analysis seriously given that you are reducing very large and complex collections of weights and data (generally hundred of megabytes of networks, plus gigabytes of images) into dendrograms and x-y plots.`
> - In our opinion, the x-y plots relating accuracy to invariance/equivariance are reasonable ways to illustrate the importance of invariance since they relate one global metric (performance) with another (overall invariance as estimated by average diameters). Also, the dendrograms are used primarily to help visualize qualitatively the similarity between SSL techniques implied by our MGMs.
> - As mentioned in our paper, the metrics have a distribution (in fact we present this distribution in Fig.3 for ViT vs ResNet). We reduce to the first, and second moment of the distribution to allow for statistical analysis and visualization across different models and tasks.
>
> `Slink is a nonparametric clustering approach, but the linkages are across different algorithms represented by MGMs. Again, this leaves the reader wondering what the significance of the linkages is---beyond a degree (distance in the MGM space) or a linkage (the closest neighbor under the slink algorithm used here).`
> - The point of clustering is to show that even SSL models with similar loss objectives/paradigms can produce largely different embedding geometry. A clear example of this is Moco-v1 and Moco-v2 models. In Table 3. included as part of the supplementary materials, we present design choices in SSL models that affect the geometric similarities between models.
> - The purpose of the clustering is to illustrate how MGMs allow us to group  SSL models.  This leads to the observation that SSL methods with similar training approaches may not always lead to similar geometry. Further, the dendrogram split based on geometry aligns with the transfer performance which is indicative of the role of geometry in transfer.
>
> `To improve the organization a table of the MGM descriptions, equations, and method of calculation would better organize the paper and make for easier digestion.`
> - We have included a table detailing MGMs in the appendix (Table 3).

---

> ### Author Response · Authors · 2022-11-19
> **Response to Reviewer HJu3 - Generalizabilty of metrics**
>
>
> `The value of the metrics is not entirely obvious, beyond the analysis provided in the paper. The conclusion that their metrics motivate the design of new SSL approaches begs the question: why did they not test this conclusion? The paper offers no concrete evidence that these metrics can be used directly for self-supervision. Furthermore, it is unclear to me why the authors limited the application to self-supervised learning. Why would these geometric features not be useful for other models? E.g. as regularization for semi-supervised models or even supervised models.`
> - In our work, we opted not to focus on a specific transfer learning task and the design of an SSL model for that task. Rather we set out to provide a general picture of how the transfer learning problem could be approached from the perspective of the geometry of pretrained SSL models. Leveraging our metric together with known regularization schemes to improve the transfer learning capability of an SSL model is in our current work agenda.
> - Given the number of self-supervised learning models available and their success, we thought it was more important to develop analysis and selection mechanisms, rather than to provide yet another SSL model. For example,  given a downstream task and multiple SSL models, one can leverage our framework to identify which model is better suited for use for the task. Note that prior to our work the only metric that was proposed for a similar use was ImageNet accuracy (CVPR 2021 [12]), the disadvantages of which are noted in our introduction.
> - We agree that the proposed framework should not be limited to an evaluation in SSL models. Our proposed metrics are general and are applicable to a broad range of settings - embeddings from deep learning models and otherwise. In this work, we focused on the application of the metrics to SSL models because, given the popularity and success of SSL, this seemed a promising first domain to test our ideas.
> We focus on SSL and analyze their geometrical properties for transfer learning as they are the state-of-the-art approach for most transfer learning tasks. It was not practically feasible within the scope of this paper to consider a wider range of deep network tasks.
>
> `For instance, why should the properties studied here be isotropic? They could certainly vary throughout the input space. This seems to limit the value of the uniform analysis applied for the entire network as if the metrics were not spatially varying in the input space(since the averages and spread are computed over a large collection of samples). The implication is that these are uniform behaviors across input space, but this is clearly not the case (imagine images from different domains with varying complexity). `
> - As discussed in the paper we use the spread of the distribution to account for the variation of the metric in the embedding space. If the metric is uniform then the spread would be zero. The discussion around ViT vs ResNet explicitly discusses this point and how we handle this.
> - The use of additional moments to capture the distribution can be done. However, we did not explore this direction.
>
> `The authors did not include a reproducibility statement in their work, however, assuming they used open source models/training code for the SSL methods here reproducing their work should be possible by a motivated scientist. `
> - The code for all our experiments and pointers to where the pretrained models were obtained is provided in the supplementary material.

---

### Official Review · Reviewer_w8qW · 2022-10-23

**Confidence:** 3
**Correctness:** 3
**Technical Novelty And Significance:** 3
**Empirical Novelty And Significance:** 3
**Recommendation:** 6

**Clarity, Quality, Novelty And Reproducibility:**

The presentation is quite clear. The empirical results are interesting.


**Strength And Weaknesses:**

#strength: It provides new metrics for evaluating the representations learned vis SSL with geometrical insights on feature similarities.
The work empirically demonstrates the effectiveness of these metrics on various SSL models and encoder network architectures.

#weakness: The current study focuses on existing network architectures and SSL paradigms. It is unclear how the metrics can lead to better SSL paradigms or representation designing methods, with better representation qualities.


**Summary Of The Paper:**

This work empirically showed that the geometry of SSL models can be efficiently captured by leveraging graph-based metrics. The work demonstrated that the proposed geometrical metrics are able to capture the transfer learning capability of many different SSL models. The analysis provides insights into the landscape of SSL algorithms, highlighting their geometrical similarities and differences.

**Summary Of The Review:**

Overall, this work is well written and well presented.
It provides new metrics for evaluating the representations learned vis SSL with geometrical insights on feature similarities.

---

> ### Author Response · Authors · 2022-11-19
> **Response to Reviewer w8qW**
>
> We appreciate the reviewer's feedback on our contribution and interest in the results.
>
>
> Note that our proposed framework can be applied to any model/architecture and thus its most immediate practical application would be, given a downstream task and multiple SSL models, to identify which model is better suited for use in that particular downstream task. Note that prior to our work, the only metric that was proposed for a similar use was ImageNet accuracy (CVPR 2021 [12]), the disadvantages of which are noted in the introduction of our paper.
>
> Also, as suggested by the reviewer, in future work, we plan to study the problem of “closing the loop” by using our proposed invariance metrics to develop better architectures. In that respect, one advantage of our proposed method is that it allows us to compare different architectures since it is based on observing how each architecture affects the data geometry. However, it is in general difficult to link properties to specific architectural and loss functions, since other factors such as normalization, batch size, number of epochs play a role. Thus, in future work, we plan to use our proposed metrics for parameter optimization given a specific architecture.

---

### Official Review · Reviewer_teRg · 2022-10-24

**Confidence:** 3
**Correctness:** 3
**Technical Novelty And Significance:** 4
**Empirical Novelty And Significance:** 4
**Recommendation:** 6

**Clarity, Quality, Novelty And Reproducibility:**

Well-written contribution.  Novel descriptors that combine manifold measures (local dimension, curvature etc). Detailed experimental analysis in supplement.

**Strength And Weaknesses:**

Strengths:
A set of intuitive descriptors "manifold graph metrics" is proposed and evidence is presented that the can inform pre-training choice for given downstream tasks.
Experimental evidence is quite impressive across imagery and audio signals

Weakness:
Some unclear points in the statistical analysis of experiments (figure 6., control for multiple comparisons?)

**Summary Of The Paper:**

The core questions in the paper are
1) What is the geometry of representations learned by contrastive learning with data augmentation?
2) What are the determinants of good generalization in downstream tasks?

**Summary Of The Review:**

An interesting meta-learning analysis with new intuitive descriptors, evidence is presented that they can inform choices in self-supervised learning

---

> ### Author Response · Authors · 2022-11-19
> **Response to Reviewer teRg**
>
> We thank the reviewer for their interest in our paper and appreciation of the novelty, presentation.
>
> We are not sure about the review comment on the statistical analysis. We observe significance values on the correlation coefficient between a transfer task and one observed MGM metric. In Figure 5 we test on metrics we consider to be crucial while in Figure 6  we study the relationship based on metrics picked as important by a LASSO regression model. We do not perform a correlation analysis of one output variable with respect to multiple input variables.

---

### Decision · Program_Chairs · 2023-01-20

**Decision:**

Reject

**Justification For Why Not Higher Score:**

After reading this paper and the comments below, I have the same concerns as Reviewer HJu3 and Reviewer w8qW did --- the proposed MGMs seem to be applicable for arbitrary representation models/paradigms, while the authors only consider the SSL methods.

Given the clustering result of various SSL methods, e.g., Figure 1, we still lack insights or evidence to improve current SSL methods. As an important question, how to use MGMs as evaluation measurements or objectives of SSL is not fully discussed. Additionally, correlations shown in the paper (e.g., the second row of Figure 6) are too far-fetched.

Furthermore, as shown in the paper, the MGMs capture the geometry of the manifold derived by SSL, but the geometry is influenced by two factors --- model architectures and learning strategies. The SSL methods apply different settings and augmentation methods (as shown in Table 4) in this work. As a result, the two factors are coupled with each other. It would be nice if the authors could decouple the factors in their analytic experiments.

**Justification For Why Not Lower Score:**

N/A

**Metareview: Summary, Strengths And Weaknesses:**

In this paper, the authors studied the geometry of self-supervised learning models. They tried to build a connection between the geometry of the learned manifold and the transferability of the model. Basically, this is an interesting analytic work that studies an important problem of self-supervised learning. However, the rationality of the proposed method should be further discussed, and the insights claimed by the authors are not very convincing.

Strengths:
(1) The motivation is clear, and the topic is interesting and important.

Weaknesses:
(1) The work is analytic, but the usefulness of its conclusion is not fully verified --- given the analytic result provided by this paper, we still lack insights or evidence to improve current SSL methods.
(2) AC and some reviewers have concerns about the rationality of the proposed MGM method. It seems too general and can be applied to analyze arbitrary learning paradigms. The reason for focusing on SSL is not explained well.